# RFMPose: Generative Category-level Object Pose Estimation via Riemannian Flow Matching

**Wenzhe Ouyang[1], Jinghua Wang[2], Zenglin Xu[3,4], Jiming Chen[1], Qi Ye[1]***
[1] Zhejiang University, [2] Harbin Institute of Technology, Shenzhen,
[3] Fudan University, [4] Shanghai Academy of AI for Science

## Abstract

We introduce RFMPose, a novel generative framework for category-level 6D object pose estimation that learns deterministic pose trajectories through Riemannian Flow Matching (RFM). Existing discriminative approaches struggle with multi-hypothesis predictions (e.g., symmetry ambiguities) and often require specialized network architectures. RFMPose advances this paradigm through three key innovations: (1) Ensuring geometric consistency via geodesic interpolation on Riemannian manifolds combined with bi-invariant metric constraints; (2) Alleviating symmetry-induced ambiguities through Riemannian Optimal Transport for probability mass redistribution without ad-hoc design; (3) Enabling end-to-end likelihood estimation through Hutchinson trace approximation, thereby eliminating auxiliary model dependencies. Extensive experiments on the Omni6DPose demonstrate state-of-the-art performance of the proposed method, with significant improvements of **+4.1** in **IoU$_{25}$** and **+2.4** in **5°2cm** metrics compared to prior generative approaches. Furthermore, the proposed RFM framework exhibits robust sim-to-real transfer capabilities and facilitates pose tracking extensions with minimal architectural adaptation. Code is available at https://github.com/shabiouyang/RMFPose.

## 1 Introduction

6D object pose estimation, which entails predicting the 3D rotation $R \in \mathrm{SO}(3)$ and 3D translation $t \in \mathbb{R}^3$ of observed objects, stands as a fundamental yet pivotal task within computer vision due to its diverse applications in augmented reality [26, 32], robotic manipulation [4, 24] and hand-object interaction [22, 29], etc. Prior works have predominantly focused on instance-level object pose estimation methods [15, 21, 12]. Although these methods, particularly recent progress [38] empowered by the Large Language Models(LLMs), have demonstrated promising performance, instance-level object pose estimation methods still suffer from limited generalization capabilities stemming from the dependency on the 3D models or RGB images for each instance. To address these limitations, category-level object pose estimation has garnered considerable attention for its generalization advantages, which eliminates the need for instance-level 3D models or RGB images during both the training and inference phases.

Existing category-level methods [36, 20, 27, 33, 10, 28] can be categorized into two distinct groups: **the correspondence-based methods** and **the direct regression-based methods**. The former approaches [36, 20, 37, 28] aim to extract features from the camera coordinate space and subsequently establish correspondences within a predefined category-specific canonical templates, including 3D NOCS [36], key-points [20], or implicit 3D embeddings [37]. However, these methods often encounter difficulties due to the non-differentiable nature of the correspondence process. In contrast,

---

*Corresponding author: Qi Ye (qi.ye@zju.edu.cn). This work was supported in part by NSFC under Grants (No.62233013, 62088101, 62293511, 62172285), Key Research and Development Program of Zhejiang Province (No.2025C01064) and Shenzhen Science and Technology Program (Project No.GXWD 20231130125451001).

the latter approaches [9, 18, 19, 10] strive to directly regress the 6D pose in an end-to-end manner. These approaches mainly focus on learning pose-sensitive features and various specialized networks, such as 3D Graph Convolution [9] and Spherical convolutions [19, 10], leveraging for the learning.

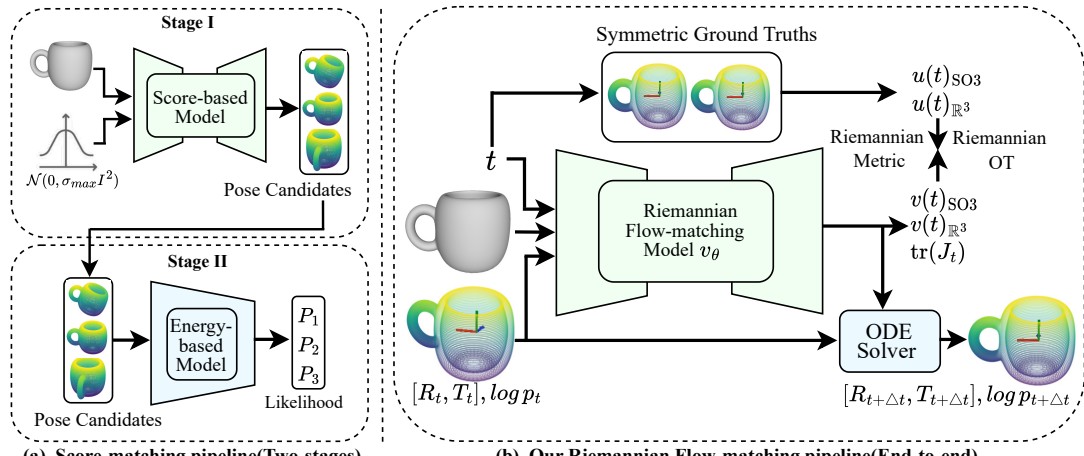

Figure 1: **Existing Score matching pipeline and our proposed Riemanian Flow Matching pipeline.** The Score matching pipeline proposed in [43] employs a two-stage framework: the first stage generates pose candidates, while the second stage estimates likelihood scores for these candidates. In contrast, our Riemannian Flow matching method models the object pose probability distribution on Riemannian manifolds to ensure geometric consistency, and simultaneously enables end-to-end likelihood estimation via trace estimation $\text{tr}(J_t)$. Moreover, our approach leverages Riemannian Optimal Transport to address the challenge of multiple feasible discrete poses induced by object symmetry. Continuous pose evolution from $t$ to $t + \triangle t$ is governed by the learned velocity field $v_\theta$ in Riemannian space with the aid of an ODE solver.

While the aforementioned methodologies have demonstrated efficacy, their fundamental conceptualization remains anchored in the discriminative paradigm, thereby inheriting two cardinal limitations: 1) difficulties in resolving the multi-hypothesis prediction problem(e.g., symmetry-induced pose multiplicity) and 2) reliance on tailored pose-sensitive feature extraction networks. Particularly regarding the second limitation, this constraint substantially hinders the flexibility of integration into thriving Vision-Language-Action (VLA) models [16, 2] for robot learning applications. To circumvent the limitations above, we advocate embracing the probabilistic methods in pose estimation, which inherently accommodates multi-hypothesis problem through probabilistic modeling while offering architectural flexibility in network design. As a seminal attempt, GenPose [43] utilized a score matching framework [31] to learn the distributions of 6D pose. However, due to the computational intractability of normalization constants in high-dimensional domains, GenPose [43] requires auxiliary training of an Energy-based model to estimate the likelihood of generated samples, as shown in Fig. 1(a). This two-stage framework inevitably introduces model complexity and sacrifices the simplicity of end-to-end training. Furthermore, score matching estimates the score function of a single-sample distribution via gradient approximation, which struggles to address the multi-target optimization in pose estimation caused by object symmetry.

To address the limitations, in this paper, we present a novel geometrically consistent framework that learns deterministic pose trajectories on Riemannian manifolds for category-level object pose estimation, termed RFMPose. The proposed RFMPose directly learns pose trajectories through Probability Flow ODEs derived from the continuity equation, which regulate probability density evolution. Our RFM framework rigorously preserves geometric constraints via two key mechanisms: (1) Geodesic-based interpolation on SO(3) via Lie algebra transformations for rotations, coupled with Euclidean interpolation in $\mathbb{R}^3$ for translations; (2) A bi-invariant Riemannian metric combining the Killing form on SO(3) with Euclidean distances in $\mathbb{R}^3$. By coalescing these components, our RFM framework guarantees physically plausible pose evolution in the SE(3) manifold.

Furthermore, we specifically address two critical challenges identified in prior works: **1) effective likelihood estimation for generative models** and **2) multi-hypothesis predictions from object symmetries**. To eliminate the requirement for auxiliary energy networks, we introduce an efficient likelihood estimation strategy for the RFM framework using Hutchinson trace estimation, thereby

enabling efficient divergence computation and end-to-end training. For symmetry-induced pose multiplicity, we propose a Riemannian Optimal Transport formulation, which minimizes the weighted geodesic cost while facilitating adaptive redistribution of probability mass across equivalent poses, as shown in Fig. 2(b). This manifold-based geometric approach resolves ambiguities by exploiting the first principles of manifold geometry, instead of relying on symmetry-specific network architectures.

Comprehensive experiments on the challenging Omni6DPose dataset demonstrate the proposed method's superiority, outperforming previous generative approaches by **+4.1** in **$IoU_{25}$** and **+2.4** in **5°2cm**. Additionally, the proposed method also achieves **42.1** in **$IoU_{25}$** under real-world scenarios without domain adaptation, verifying its inherent sim-to-real transfer capability. Moreover, the proposed method permits direct extension to object pose tracking through marginal architectural adjustments and demonstrates competitive performance accuracy on object pose tracking.

The principal contributions of this work can be summarized as follows:

- We establish a Riemannian Flow matching framework that leverages Riemannian interpolation and metric to ensure manifold-consistent trajectory learning for 6D pose estimation;

- We propose end-to-end likelihood estimation with the Hutchinson trace estimation on 6D pose estimation, which eliminates the requirement for auxiliary models.

- We design Riemannian Optimal Transport to resolve symmetry-induced pose multiplicity in 6D pose estimation.

- Extensive experiments on the challenging Omni6DPose dataset verify the superior performance of the proposed method compared to state-of-the-art approaches and demonstrate the great potential of the proposed RFM framework in object pose estimation.

## 2   Related Works

**Correspondence-based Category-level Pose Estimation.** This family of methodologies [36, 20, 37, 28] seeks to establish the correspondence between camera coordinate space and the predefined category-specific canonical templates, subsequently recovering poses via optimization-based fitting algorithms (e.g., Umeyama alignment [35]). As a seminal advancement, NOCS [36] introduced a unified canonical representation to align intra-category object instances geometrically. Building upon this foundation, SpherePose [28] utilizes spherical feature interaction mechanisms to achieve enhanced correspondence precision through geodesic-aware feature matching. Besides, AG-Pose [20] advocated geometry-driven keypoint detection as an alternative correspondence paradigm, while Query6DoF [37] developed implicit shape priors through learnable sparse query matching, circumventing explicit template constraints. SAR-Net [17] and RBP-Net [44] focused on symmetry-correspondence to mitigate the symmetry-induced pose multiplicity. Notwithstanding these advancements, the correspondence process is inherently non-differentiable and cannot be integrated into the learning process. Consequently, inaccuracies in generating predefined category-specific canonical templates exert a significant influence on the accuracy of pose estimation, as error propagation remains unmitigated through gradient-based optimization.

**Direct Regression-based Category-level Pose Estimation.** This category of approaches [9, 18, 19, 10] aims to regress the object pose in an end-to-end manner directly. FS-Net [9] proposes to decouple the rotation into two perpendicular vectors, simplifying prediction, and utilizes a 3D Graph Convolution autoencoder for feature extraction. VI-Net [19] leverages spherical representations to decouple the rotation into a viewpoint rotation and an in-plane rotation, thereby simplifying the challenge of rotation estimation. Based on the decoupled representation, SecondPose [10] proposed to extract SE(3)-consistent semantic and geometric features to enhance pose estimation accuracy. However, these methods struggle with the pose-sensitive feature learning due to the non-linearity of SE(3). Furthermore, excessive reliance on specialized pose-sensitive feature extraction networks undermines model simplicity, impeding seamless integration with contemporary VLA frameworks.

**Generative Modeling for Object Pose Estimation.** Recently, generative modeling has emerged as a promising paradigm for various tasks far beyond classic generation tasks, such as classification [6], perception [39], and robotics action planning [11, 41]. As a pioneering work, GenPose [43] proposed to learn 6D pose distribution by score matching. However, score matching struggles to estimate probabilities in high-dimensional manifolds like SE(3) and fails to resolve pose ambiguity caused by object symmetry. As a comparison, flow matching [23] learning deterministic trajectories via Probability Flow ODEs. Recent advances in Riemannian manifolds [8, 3, 14] have demonstrated the

capacity of flow matching to model complex geometric transformations. In this paper, we pioneer the application of Riemannian Flow matching to category-level 6D pose estimation, systematically addressing the geometric constraints inherent in the object pose estimation.

## 3 Methodology

We will first introduce the core mechanism of learning pose distributions via Riemannian Flow Matching. Subsequently, we will detail how we address the challenge of object symmetries using Riemannian Optimal Transport. Finally, we explain our end-to-end likelihood estimation technique, which employs Hutchinson trace estimation to obviate the need for auxiliary models.

### 3.1 Preliminaries

**Problem Formulation.** The 6D pose estimation task aims to estimate 6D object pose $[R_i, T_i]$, where $R \in \mathbb{R}^{3 \times 3}$ is a rotation matrix and $T \in \mathbb{R}^3$ is a translation vector, using the given multi-modal sensory inputs: a partially observed point cloud $\mathbf{O}_i \in \mathbb{R}^{3 \times N}$ and a cropped RGB image $\mathbf{I}_i \in \mathbb{R}^{3 \times H \times W}$. Therefore, the learning agent is given a training set with a paired dataset $\mathcal{D} = ([R_i, T_i], \mathbf{O}_i, \mathbf{I}_i)_{i=1}^n$.

**Conditional Continuous Normalizing Flows for Pose Generation.** To model a target conditional distribution $q([R,T]|c)$ for a given condition variable $c$, we transform a prior conditional distribution $\rho_0([R,T]|c)$ with a velocity fields conditioned on $c$. This transformation is guided by the following Ordinary Differential Equations (ODEs):

$$\frac{d[R,T]}{dt} = v_\theta(t, c, [R,T]), \tag{1}$$

where $\theta$ are trainable parameters and $t \in [0,1]$. This ODE equation generates a flow and a conditional probability density path $\rho_t([R,T]|c)$. In this paper, the condition variable $c$ represents a partially observed point cloud $\mathbf{O}_i$ and a cropped RGB image $\mathbf{I}_i$. The target distribution $q([R,T])$ corresponds to the distribution of 6D poses of the ground truth in the datasets.

### 3.2 Learning Pose Distribution via Riemannian Flow Matching

Conditional flow paths in Eq. (1) are designed primarily under the assumption of Euclidean geometry, resulting in linear interpolations. However, this can be particularly restrictive for tasks such as trajectory inference, where straight paths might lie outside the data manifold, thus failing to capture the underlying dynamics giving rise to the observed marginals.

In this paper, we tackle the aforementioned issue by learning the 6D pose distribution within a Riemannian space, which facilitates geodesic-based interpolations using minimal-length curves under Riemannian distance [8]. Prior to delving into the method, we first delineate the Riemannian structure inherent in the 6D pose estimation task. Conventionally, a pose matrix $[R,T]$ in Euclidean space can be transformed into SE(3) manifolds. According to [3], the disintegration of measures posits that every SE(3)-invariant measure can be decomposed into an SO(3)-invariant measure and a measure proportional to the Lebesgue measure on $\mathbb{R}^3$. This enables us to simplify the construction of independent flows on SO(3) and $\mathbb{R}^3$ for simplicity. To construct a conditional vector field from $R_0$ to $R_1$ on the Riemannian space SO(3), we leverage the Lie algebra $\mathfrak{so}(3)$, which is comprised of skew-symmetric matrices acting as tangent vectors at the identity of SO(3). The geodesic interpolation at $t$ in SO(3) can be formulated as:

$$u(t)_{\mathrm{SO3}} = R(t) = R_0 \cdot \exp\left(t \cdot \log(R_0^\top R_1)\right), \tag{2}$$

where $\log : \mathrm{SE}(3) \to \mathfrak{se}(3)$ is the Lie algebra transformation, and $\exp : \mathfrak{se}(3) \to \mathrm{SE}(3)$ is the Lie group transformation. $R_0$ and $R_1$ are orthogonal rotation matrices in the initial and target states. Constructing a conditional vector field for translation on $\mathbb{R}^3$ can be simplified via Euclidean interpolation:

$$u(t)_{\mathbb{R}^3} = T(t) = (1-t)T_0 + tT_1, \tag{3}$$

where $T_0$ to $T_1$ are the translation vector in the initial state and target state. Capitalizing on the above Riemannian interpolation, we can derive the Riemannian flow matching framework for 6D pose distribution through the following formulation:

$$\mathcal{L}_{\mathrm{RCFM}}(\theta) = \mathbb{E}_{t, q([R,T]), p_t([R,T]|c)} \|v_\theta(t, c, [R,T]) - u(t, [R,T])\|_{\mathrm{SE3}}^2 \tag{4}$$

Herein, we can model the target distribution $q([R,T]|c)$ by sampling from a predefined prior distribution $\rho_0([R,T]|c)$ and evolving these initial samples along a Riemannian flow over $t \in [0,1]$, as depicted in Fig. 2.

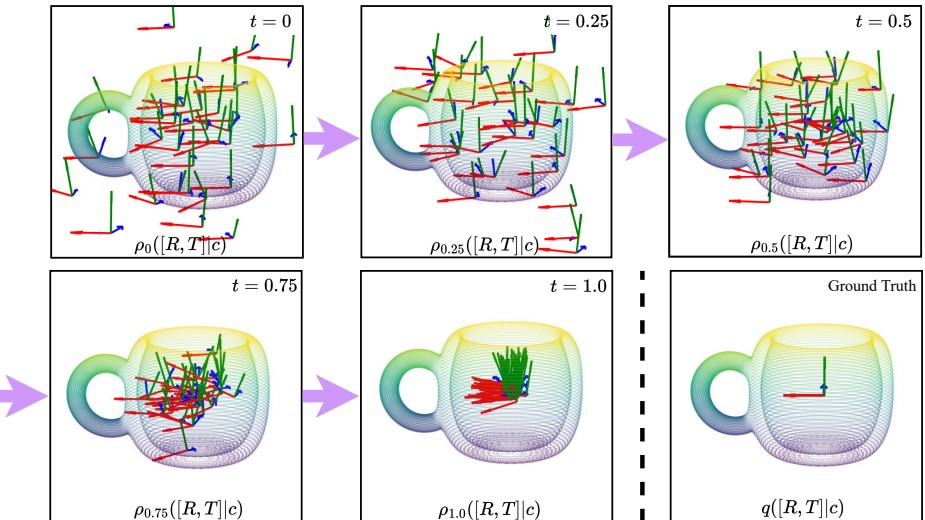

Figure 2: **The illustration of object pose generation Process with Riemannian flow matching.** We leverage geodesic interpolation on SO(3) and Euclidean interpolation in $\mathbb{R}^3$ to derive deterministic pose trajectories, ensuring geometric consistency across rotational and translational components.

### 3.3 Riemannian Optimal Transfer For Symmetry

Building upon the aforementioned Riemannian flow matching framework, we construct conditional probability paths to learn 6D pose distributions from given datasets. However, a critical challenge in 6D pose estimation lies in handling symmetric objects (e.g., bottles), where multiple feasible ground truths exist for a single object pose. In addressing the coexistence of heterogeneous asymmetric and symmetric objects in the 6D pose estimation task, we divide the construction of conditional probability paths into two distinct scenarios: single-hypothesis and multi-hypothesis.

First, consider the scenario of a single-hypothesis. The optimal transformation must map the single source pose to the single target pose via the unique shortest path. Therefore, the objective of Optimal Transport(OT) [34] in Riemannian manifolds is akin to SE(3) geodesic interpolation, as both aim to find the shortest path. For scenarios involving multiple-hypotheses, there exist multiple feasible ground truth poses. Assuming continuous source distribution $\rho_0 = \delta_{[R_0, T_0]}$ and target distribution $\rho_1 = \delta_{[R_1, T_1]}$, where $\delta_{(\cdot)}$ denotes the Dirac measures. The general form of the Riemannian OT [3] for constructing the optimal conditional probability paths is given by:

$$\mathrm{OT}(\rho_0, \rho_1) = \inf_{\Phi \in \mathcal{C}} \int_{\mathrm{SE}(3)} c\left([R_x, T_x], \Phi([R_x, T_x])\right) \rho_0([R_x, T_x]) d\mu([R_x, T_x]), \qquad (5)$$

where $\mathcal{C}$ is the set of admissible transport plans on SE(3), $\Phi([R_x, \mathbf{T}_x])$ represents the transformed pose via $\Phi(\cdot)$, and $d\mu$ is the Haar measure on SE(3). In this paper, the cost function $c([R_1, \mathbf{t}_1], [R_2, \mathbf{t}_2])$ is defined on Riemannian manifolds, more specifically, the SE(3) manifold:

$$c([R_1, \mathbf{t}_1], [R_2, \mathbf{t}_2]) = \left\| \log\left([R_1, \mathbf{t}_1]^{-1}[R_2, \mathbf{t}_2]\right) \right\|_{\mathfrak{se}(3)} \qquad (6)$$

Because the target distribution $\rho_1 = \delta_{[R_1, T_1]}$ has multiple feasible ground truth poses, $\rho_1$ can be rewritten as a discrete distribution $\rho_1 = \sum_j \beta_j \delta_{[R_{y,j}, \mathbf{T}_{y,j}]}$. Subsequently, the objective simplifies to minimizing the weighted average cost over target poses:

$$\mathrm{OT}(\rho_0, \rho_1) = \inf_{\Phi \in \mathcal{C}} \sum_j \beta_j \left\| \log\left([R_0, \mathbf{t}_0]^{-1}[R_{y,j}, \mathbf{t}_{y,j}]\right) \right\|_{\mathfrak{se}(3)}, \qquad (7)$$

where $\beta_j$ is the coefficient for $j$-th discrete distribution. In the field of 6D pose estimation, it is commonly assumed that multiple ground-truth poses are attributable to object symmetries. Consequently, these poses have equal occurrence probabilities, leading to identical coefficients $\beta$ for discrete distributions. This formulation allows the transport map $\Phi(\cdot)$ to distribute "probability mass" from the single pose to multiple poses, guided by the SE(3) Riemannian metric.

### 3.4 Likelihood Estimation for Pose Candidates

Although the Riemannian flow matching model enables conditional sampling from pose distributions, the 6D pose estimation task often necessitates a deterministic and numerically accurate output. To address this challenge, we must develop a strategy for selecting or aggregating a final output estimation from multiple generated samples. Due to the vanilla mean pooling of 6D pose samples typically leading to a significant statistical bias induced by outliers in the distribution tails, GenPose [43] trained a decoupled Energy-based model [30] that performs likelihood estimation for its generating candidates. However, this Energy-based approach deprives the model of the advantages of end-to-end training.

To achieve end-to-end training, we estimate the likelihood of generated samples in flow matching using Hutchinson trace estimation [13], which eliminates the need for auxiliary likelihood estimation models. As depicted in Fig. 1(b), the evolution of log-likelihood for Riemannian flow matching $\log p_t([R_t, T_t])$ depends on a continuous ODE equation:

$$\frac{\partial \log p_t([R_t, T_t]])}{\partial t} = -\nabla \cdot (v_\theta([R_t, T_t], t)), \tag{8}$$

where $\nabla(\cdot)$ denotes a divergence operation. The Log-likelihood can be computed by integrating $t \in [0, 1]$:

$$\log p_1(v_\theta([R_t, T_t])) - \log p_0(v_\theta([R_0, T_0])) = -\int_0^1 \nabla \cdot (v_\theta([R_t, T_t])) \, dt \tag{9}$$

To calculate the divergence of the velocity field $\nabla \cdot (v_\theta([R_t, T_t]))$ is equivalent to solving the trace of its Jacobian matrix:

$$\nabla \cdot (v_\theta([R_t, T_t], t)) = \text{tr}(J_t) = \sum_{i=1}^{D} \frac{\partial v_{t,i}([R_t, T_t], t)}{\partial x_{t,i}}, \tag{10}$$

where $\text{tr}(J_t)$ represents the trace of the Jacobian matrix. Calculating the trace of the Jacobian matrix in high-dimensional spaces involves significant computational complexity, which is infeasible for real-time applications. To tackle this issue, we utilize the Hutchinson trace estimator [13], which enables us to approximate the divergence using an unbiased estimation. Specifically, we first generate a random vector $\epsilon$ with the same dimension as $v_\theta([R_t, T_t])$, typically sampled from the standard normal distribution $\mathcal{N}(0, I)$. Then, we calculate the Jacobian-Vector Product (JVP), $J_{v_\theta([R_t, T_t])}\epsilon$, which can be efficiently calculated using automatic differentiation tools in PyTorch. Finally, we repeat the this operation $N$ times to obtain the expectation of JVP, which can be regarded as an approximate estimation of divergence:

$$\text{tr}(J_t) = \mathbb{E}[\epsilon^T J_t \epsilon] \tag{11}$$

After obtaining the integration term in Eq. (9), we still need to calculate $\log p_0(v_\theta([R_t, T_t]))$. Since the initial state $[R, T]$ follows a standard normal distribution $\mathcal{N}(0, I)$ with density function $p_0(\mathbf{x}) = \frac{1}{(2\pi)^{d/2}} \exp\left(-\frac{1}{2}\mathbf{x}^\top \mathbf{x}\right)$, the log-likelihood at the initial time step can be computed as follows:

$$\log p_0(v_\theta([R_0, T_0])) = -\frac{1}{2}[R_0, T_0]^\top [R_0, T_0] - \frac{d}{2}\log(2\pi), \tag{12}$$

where $d$ denotes the dimension of 6D pose. After acquiring the likelihood values for each pose candidate, we discard candidates with likelihoods below the threshold $\delta$. Finally, the retained candidates are then aggregated by computing the weighted average of rotations in SO(3) and translations in $\mathbb{R}^3$, respectively.

### 3.5 Discussion

**Why RFM Enables Geometric-Consistency.** Score matching aims to learn the noise in the denoising process, which inherently lacks physical meaning(orthonormalization is only applied to the final outputs as a post-processing step to ensure its physical legitimacy). In contrast, flow matching directly learns a velocity vector field $v_\theta(t, c, [R, T])$ that governs the evolution of poses, endowed with explicit physical interpretability. This enables enforcing geodesic constraints on the Riemannian manifold, thereby ensuring geometric consistency throughout the pose generation process.

**Why RFM Enables End-to-End Likelihood Estimation.** Score matching learns score functions (probability gradients) rather than the probability distribution itself, thereby suffering from the calculation of the intractable normalization constant, especially in high-dimensional spaces. In contrast, flow matching explicitly models deterministic probability flows through velocity fields derived from the continuity equation, inherently ensuring probability conservation, reversible trajectories, and stable trace computation via Jacobian determinants. Moreover, flow matching's direct optimization of velocity fields mitigates the instability of score matching in low-density regions, where score gradients become ill-defined due to sparse sampling.

## 4 Experiments

### 4.1 Experimental Setup

**Datasets.** Since our generative modeling framework does not require any category-specific canonical priors, this obviates the need for an effortless extension of the framework to datasets containing numerous object categories. Therefore, we conduct experiments on Omni6DPose [42], a novel yet challenging benchmark dataset for 6D pose estimation. This comprehensive dataset comprises 807K synthetic and real images with over 6.5 million annotations spanning 149 object categories. Notably, the diversity and scale of Omni6DPose [42] significantly surpass prevailing datasets like REAL275 [36], which contains only 7K images restricted to 6 common object categories. We train our models exclusively on synthetic data for all experiments and evaluate performance across both synthetic and real-world data.

**Implementation Details.** Following the baseline established in Omni6DPose [42], we employ RGB and point cloud modalities as dual input streams for both training and inference phases. For RGB image input, a pre-trained, frozen DINOv2 model is utilized to extract semantic feature representations. For the point cloud input, we leverage Farthest Point Sampling (FPS) to subsample 1,024 points, followed by global feature extraction via PointNet++. During the feature aggregation stage, the RGB features are spatially concatenate with corresponding point coordinates to construct cross-modal fused representations. Please refer to the Supplementary Materials for more implementation details.

### 4.2 Comparison with State-of-the-art Methods

**Results on Simulation Datasets.** We first compare the proposed method with other existing methods under simulation settings. The Omni6DPose [42] contains the synthetic data based on three classic datasets: ScanNet++ [40], IKEA [1], and Matterport3D [5]. Table 1 presents comparative evaluations of the proposed method against state-of-the-art methods on the Omni6DPose ScanNet++ test-set. As shown in Table 1, our approach surpasses all deterministic methods by a large margin across all evaluation metrics, which demonstrates the potential of conditional generative modeling for category-level object pose estimation. Notably, even when compared with the state-of-the-art generative method GenPose++ [42], our solution maintains a significant performance advantage. Specifically, the proposed method leads by over **+4.1** in $IoU_{25}$ and **+3.4** in **5°2cm**.

**Results on Real-world Datasets.** To further validate the efficacy of our approach, we also evaluate our approach on real-world datasets. Notably, we still train our models exclusively on the aforementioned synthetic data. Table 2 shows the comparison of our method with state-of-the-art methods on the Omni6DPose ROPE set. As shown in Table 2, the proposed method significantly outperforms existing solutions across all quantitative metrics, demonstrating the sim-to-real transfer capability of our

Table 1: **Quantitative comparison of category-level object pose estimation on Omni6DPose ScanNet++ test-set.** The results are averaged over all 149 categories.

| Method | End-to-End Training | Input Modality | IoU | | | AUC | | | |
|---|---|---|---|---|---|---|---|---|---|
| | | | $IoU_{25}$ | $IoU_{50}$ | $IoU_{75}$ | 5°2cm | 5°5cm | 10°2cm | 10°5cm |
| **Deterministic:** | | | | | | | | | |
| - HS-Pose [45] | ✓ | Point Clouds | 31.1 | 12.0 | 1.7 | 3.4 | 6.1 | 7.9 | 13.4 |
| - AG-Pose [20] | ✓ | RGB-D | 29.9 | 10.6 | 1.1 | 2.2 | 4.3 | 6.2 | 10.1 |
| - SecondPose [10] | ✓ | RGB-D | 31.5 | 12.2 | 2.0 | 3.1 | 7.9 | 11.3 | 16.7 |
| **Probabilistic:** | | | | | | | | | |
| - GenPose++ [42] | ✗ | RGB-D | 43.9 | 24.7 | 3.3 | 10.4 | 13.2 | 21.7 | 28.5 |
| - **Ours** | ✓ | RGB-D | **48.0** | **28.9** | **5.0** | **12.8** | **16.2** | **25.2** | **31.6** |

Table 2: **Quantitative comparison of category-level object pose estimation on Omni6DPose ROPE set**. The results are averaged over all 149 categories.

| Method | Input Modality | IoU | | | AUC | | | |
|---|---|---|---|---|---|---|---|---|
| | | $\text{IoU}_{25}$ | $\text{IoU}_{50}$ | $\text{IoU}_{75}$ | 5°2cm | 5°5cm | 10°2cm | 10°5cm |
| **Deterministic:** | | | | | | | | |
| - NOCS [36] | RGB-D | 0.0 | 0.0 | 0.0 | 0.0 | 0.0 | 0.0 | 0.0 |
| - SGPA [7] | RGB-D | 10.5 | 2.0 | 0.0 | 4.3 | 6.7 | 9.3 | 15.0 |
| - IST-Net [25] | RGB-D | 28.7 | 10.6 | 0.5 | 2.0 | 3.4 | 5.3 | 8.8 |
| - HS-Pose [45] | Point Clouds | 31.6 | 13.6 | 1.1 | 3.5 | 5.3 | 8.4 | 12.7 |
| - AG-Pose [20] | RGB-D | 29.3 | 10.9 | 0.7 | 2.1 | 3.5 | 6.7 | 9.2 |
| - SecondPose [10] | RGB-D | 33.6 | 15.4 | 2.0 | 5.0 | 7.3 | 10.4 | 15.1 |
| **Probabilistic:** | | | | | | | | |
| - GenPose [43] | Point Clouds | - | - | - | 6.6 | 9.6 | 13.1 | 19.3 |
| - GenPose++ [42] | RGB-D | 39.0 | 19.1 | 2.0 | 10.0 | 15.1 | 19.5 | 29.4 |
| - **Ours** | RGB-D | **42.1** | **21.0** | **2.2** | **10.4** | **15.7** | **21.0** | **30.8** |

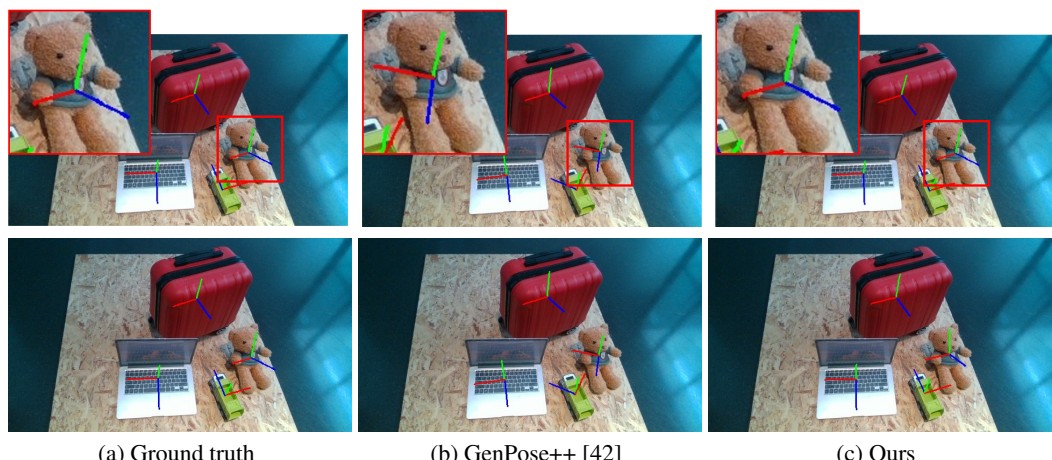

(a) Ground truth        (b) GenPose++ [42]        (c) Ours

Figure 3: **Visualization comparison on Omni6DPose [42]**. As shown in the zoomed area of the figure above, our approach has achieved better performance than GenPose++ [42].

proposed Riemannian flow matching framework. Figure 3 presents detailed comparative visualization results of our model against GenPose++ [42] and the ground truths.

**Results on Category-level Object Pose Tracking.** The closed-loop generative architecture inherent in the flow matching framework enables seamless adaptation of the proposed method to the object pose tracking task with minimal modification. Technically, we perturb the pose input $R_t T_t$ of ODE Solver in Fig. 2 with a Gaussian noise, while initializing the input $t$ as $t_\delta \in (0, 1)$. By default, we set the $t_\delta = 0.55$ in this paper. Moreover, we employ the same likelihood estimation and aggregation strategy with a single-frame pose estimation framework to obtain the estimation of the current frame. The comparison of category-level object pose tracking on the Omni6DPose ROPE set is presented in Table 3. As demonstrated in Table 3, our method maintains a leading position in the object pose tracking task. Notably, the proposed Riemannian flow matching framework not only enables end-to-end training but also offers faster inference speed compared to GenPose++ [42].

Table 3: **Comparison of category-level object pose tracking on Omni6DPose ROPE.** The results are averaged over all 149 categories.

| Method | Input | FPS↑ | 5°5cm ↑ | mIoU↑ | $R_{err}(°) \downarrow$ | $T_{err}$(cm)↑ |
|---|---|---|---|---|---|---|
| - GenPose [43] | Point Clouds | **11.7** | 13.3 | - | 19.3 | **1.2** |
| - GenPose++ [42] | RGB-D | 8.7 | 15.9 | 53.4 | 17.6 | **1.2** |
| - **Ours** | RGB-D | 11.3 | **16.1** | **54.1** | **15.9** | **1.2** |

Table 4: Ablation studies on the Riemannian Interpolation and Metric.

| Ablation | $IoU_{25}\uparrow$ | $5°2cm\uparrow$ | $5°5cm\uparrow$ |
|---|---|---|---|
| Vanilla Euclidean Interp. | 43.4 | 9.7 | 13.9 |
| + Riemannian Interp. | 45.1(**+1.7**) | 10.4(**+0.7**) | 14.7(**+0.8**) |
| **++ Riemannian Metric** | **48.0(+4.6)** | **12.8(+3.1)** | **16.2(+2.3)** |

Table 5: Ablation studies on the Likelihood Estimation and Samples Aggregation.

| Ablation | $IoU_{25}\uparrow$ | $5°2cm\uparrow$ | $5°5cm\uparrow$ |
|---|---|---|---|
| Maximum likelihood | 29.7 | 6.6 | 9.3 |
| with Averaging | 46.8 | 10.8 | 15.5 |
| **Weighted Averaging** | **48.0(+1.2)** | **12.8(+2.0)** | **16.2(+0.7)** |

## 4.3 Ablation Studies

We conduct ablation studies on the Scannet++ test set of Omni6DPose [42] from three perspectives: (1) the effectiveness of Riemannian interpolation and metric; (2) the impact of likelihood estimation and sample aggregation; (3) the role of Riemannian OT for symmetric objects.

**Effectiveness of the Riemannian Interpolation and Metric.** In this paper, we introduce Riemannian interpolation and Riemannian metric to enable more efficient modeling of and learning from 6D pose distributions. To this end, we first conduct an ablation study on the roles of these two core components. Table 4 presents ablation results for the Riemannian interpolation and metric. As shown in Table 4, the Riemannian interpolation design effectively improves performance in $IoU_{25}$ by **+1.7** and **$5°2cm$** by **+0.7**. The Riemannian Metric further boosts he 6D pose estimation performance by **+4.6** in $IoU_{25}$ and **+3.1** in **$5°2cm$**.

**Ablation studies on the Likelihood Estimation and Samples Aggregation.** Table 5 presents ablation results for the proposed likelihood estimation method and different sample aggregation strategies. As shown in Table 5, the multiple sample aggregation strategy (2nd and 3rd columns) surpasses the single sample obtained by maximum likelihood estimation by a large margin, verifying the superiority of generative models in reducing pose estimation error through multiple samplings. Moreover, the weighted averaging strategy outperforms standard averaging with improvement **+1.2** in $IoU_{25}$ and **+2.0** in **$5°2cm$**, validating the effectiveness of the proposed likelihood estimation method.

**Effectiveness of the Riemannian OT for Symmetric Objects.** In this paper, we introduce Riemannian OT to address the challenge posed by multiple feasible poses of symmetric objects in 6D pose estimation. To experimentally validate the effectiveness of Riemannian OT, we incorporated the half-symmetric property into

Table 6: Ablation studies on the Riemannian Optimal Transfer(ROT) for Symmetric Objects.

| Method | $IoU_{25}\uparrow$ | $5°2cm\uparrow$ | $5°5cm\uparrow$ |
|---|---|---|---|
| GenPose++ [42](Symmetric) | 41.5 | 11.1 | 12.5 |
| Ours(w/o ROT)(Symmetric) | 43.4 | 12.1 | 15.2 |
| Ours(with ROT)(Symmetric) | 48.5(**+5.1**) | 13.7(**+1.6**) | 17.1(**+1.9**) |

Omni6DPose [42] for comparative experiments. As demonstrated in Table 6, Riemannian OT successfully alleviates this issue, leading to significant performance improvements compared to configurations without it. Notably, the proposed method is independent of symmetry-specific network architectures or custom loss designs.

## 5 Conclusion

In this paper, we present the Riemannian Flow Matching (RFM) for category-level 6D pose estimation, which learns deterministic pose trajectories via geodesic interpolations while explicitly preserving geometric constraints. The key contributions of our work are threefold: 1) a Riemannian manifold-based probabilistic path modeling for 6D pose estimation; 2) probability mass redistribution for symmetry-induced pose multiplicity through Riemannian Optimal Transport; 3) an efficient likelihood estimation strategy with trace estimation for end-to-end training. Comprehensive evaluations on the challenging Omni6DPose dataset demonstrate that RFM significantly outperforms state-of-the-art baselines. With its simple architecture and compatibility with advanced generative models, our approach offers a robust foundation for integrating into unified robot learning frameworks.

**Limitations and Future Works:** Although our RFM model has achieved promising performance, its accuracy remains unsatisfactory on articulated objects (e.g., laptops). Given the prevalence of such articulated objects in hand-object interactions, our future work will focus on two key aspects: 1)Enhancing the RFM framework to improve pose estimation accuracy for articulated objects; 2) exploring integration of the RFM framework into emerging vision-language-action (VLA) models, enabling end-to-end perception-to-manipulation pipelines.

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
