# OpenReview forum: "RFMPose: Generative Category-level Object Pose Estimation via Riemannian Flow Matching"
_NeurIPS.cc/2025/Conference — NeurIPS 2025 poster_

### Official Review · Reviewer_iY4w · 2025-06-10

**Clarity:** 2
**Significance:** 3
**Originality:** 3
**Rating:** 4
**Confidence:** 4

**Summary:**

This work applies conditional flow matching (CFM) to the category-level pose estimation task. Specifically, it first formulate category-level pose estimation as a conditional probability distribution fitting problem and derive the corresponding CFM loss. Second, pose distance is measured in Riemannian space (as opposed to traditional Euler space) to construct the conditional vector field, while incorporating symmetry-aware pose distance metrics for optimal transport (OT) computation on Riemannian manifolds. Finally, the category-level pose is obtained by weighting multiple generated pose candidates based on their likelihood values.

**Questions:**

See Weaknesses.

**Ethical Concerns:**

["NO or VERY MINOR ethics concerns only"]

**Final Justification:**

The authors provided reasonable explainations of my concerns, and supplemented necessary ablations. I'm glad to hold my borderline accept for this paper.

**Limitations:**

Yes.

**Paper Formatting Concerns:**

NA.

**Quality:**

3

**Strengths And Weaknesses:**

Strengths:

1. Theoretical Contribution: The work leverages the advantages of flow matching to address limitations of score matching (e.g., in GenPose), with rigorous derivations. The integration of Riemannian geometry for pose distance/interpolation with flow matching is innovative and well-motivated.
2. Experimental Validation: Comprehensive comparative experiments substantiate the effectiveness of the proposed contributions.

Weaknesses:

1. Marginal Gains: Although the utilization of the flow model and the analysis of Riemannian geometry are impressive, the improvements on the real-world dataset (ROPE) are relatively modest. I'd like to know more about insights on it.
2. More Ablations: There's no discussion or ablation study on the impact of the number of pose candidates on performance, which should be very important to assess a generative model.
3. Efficiency: What's the inference time of the proposed method? It's important for industrial applications.
3. Writing Issues: The framework lacks a clear schematic diagram, which would aid in understanding the overall pipeline.
Line 158: Incorrect explanation of log and exp operations (should specify SO(3)).
Line 221: Redundant phrase "the this".

---

> ### Author Rebuttal · Authors · 2025-07-31
>
> Dear Reviewer:
>
> We sincerely appreciate your acknowledgement of our work, particularly your recognition that our idea is **"rigorous derivations"** and **"innovative and well-motivated"**. We hope to address all your concerns with the following responses:
> _________________
> **Q1. Marginal Gains: Although the utilization of the flow model and the analysis of Riemannian geometry are impressive, the improvements on the real-world dataset (ROPE) are relatively modest. I'd like to know more about insights on it.**
>
> **A1:** Thank you for your insightful comment. We agree that the performance gains on the real-world ROPE dataset are more modest compared to those on the synthetic dataset, and we appreciate the opportunity to elaborate on the reasons.
>
> **First**, we would like to emphasize a crucial aspect of our experimental setup: all models are trained exclusively on the synthetic (SOPE/ScanNet++) dataset and evaluated on both simulation and real-world datasets. This poses a highly challenging zero-shot Sim2Real transfer task. The well-known domain gap between synthetic and real-world data, stemming from differences in visual appearance (e.g., textures, lighting), sensor noise, and scene complexity, creates a significant performance bottleneck.
>
> **Second**, a closer analysis of the metrics in Table 1 and Table 2 reveals a telling pattern. Compared to GenPose++[1], our method achieves more substantial improvements on the ROPE dataset on metrics with higher tolerance, such as $\mathbf{IoU}_{25}$(**+3.1**), $10^∘2$cm(**+1.5**), and $10^∘5$cm (**+1.4**). However, the gains are indeed more modest on the most stringent high-precision metrics, namely $5^∘2$cm(**+0.4**) and $5^∘5$cm (**+0.6**).
>
> We posit that this pattern directly reflects the impact of the Sim2Real gap. The noise and appearance shifts from the real world likely introduce a "performance floor" on the final achievable precision. Our method, by leveraging flow models and Riemannian geometry, learns a more accurate pose distribution than GenPose++. This advantage is fully realized on clean, in-domain synthetic data. When confronted with real-world data, while this precision advantage is partially masked by the noise induced by the domain gap, the superior geometric priors learned by our model still lead to a more accurate and robust pose estimate overall.  This is clearly demonstrated by the more significant gains on the less strict metrics (such as $\mathbf{IoU}_{25}$).
>
> In summary, we believe the more modest, yet still consistent and comprehensive, gains on the ROPE dataset should not be seen as a limitation of our model's theoretical foundation, but rather as a testament to its successful generalization in a demanding zero-shot Sim2Real setting. The results demonstrate that our approach consistently outperforms the strong baseline even when challenged by a significant domain gap. We believe these findings are very promising and represent a solid step towards solving category-level pose estimation in the real-world.
>
> [1] Omni6DPose: A Benchmark and Model for Universal 6D Object Pose Estimation and Tracking, ECCV 2024.
> _________________
> **Q2. More Ablations: There's no discussion or ablation study on the impact of the number of pose candidates on performance, which should be very important to assess a generative model.**
>
> **A2:** Thanks for your insightful comment. Below is the ablation study on the impact of the number of pose candidates on performance. As shown in the table, when the number of pose candidates increases from 10 to 50, the performance improvement is significant, but when it increases from 50 to 100, the performance improvement is already very marginal, while the computational resources required for inference still increase.
>
> | Number of pose candidates | $\mathbf{IoU}_{25}$ | $\mathbf{IoU}_{50}$ | $\mathbf{IoU}_{75}$ | $5^{∘}2cm$ | $5^{∘}5cm$ | $10^{∘}2cm$ | $10^{∘}5cm$ |
> | :---: | --- | --- | --- | --- | --- | --- | --- |
> | 10 | 44.6 | 21.1 | 3.9 | 11.4 | 15.1 | 21.7 | 29.3 |
> | 30 | 46.8 | 27.9 | 4.6 | 12.2 | 15.7 | 24.6 | 30.1 |
> | 50 | 48.0 | 28.9 | 5.0 | 12.8 | 16.2 | 25.2 | 31.6 |
> | 100 | 48.2 | 29.1 | 6.0 | 12.8 | 16.4 | 25.8 | 31.9 |
> _________________
> **Q3. Efficiency: What's the inference time of the proposed method? It's important for industrial applications.**
>
> **A3:** Thank you for your valuable comment regarding the inference time of our proposed method. We completely agree that inference efficiency is a crucial aspect for industrial applications.
>
> We have reported the inference speed of our proposed method in Table 3 of the manuscript, measured in **Frames Per Second (FPS)**. As shown in the Table 3, our model achieves 11.3 FPS on a single A100 GPU (under equivalent computing power). We appreciate your attention to this key performance metric. We hope the results in Table 3 adequately addresses your question about the model's efficiency.
> _________________
> **Q4. Writing Issues: The framework lacks a clear schematic diagram, which would aid in understanding the overall pipeline. Line 158: Incorrect explanation of log and exp operations (should specify SO(3)). Line 221: Redundant phrase "the this".**
>
> **A4:** Thanks for your kind suggestion. Figure 2 illustrates the overall process of Riemannian Flow Matching for pose estimation, showing the learning process from a random Gaussian pose to the target pose.  Figure 1(b) details the step-by-step process within the RFM framework, including velocity field matching, likelihood estimation, and supervision. We will update our figure and add a paragraph at the beginning of Section 3 in revised manuscript to further clarify the overall pipeline of the proposed method.
>
> We thank the reviewer for pointing out these typos, and we have corrected them in the revised manuscript.

---

> > ### Comment · Reviewer_iY4w · 2025-08-07
> >
> > The authors provided reasonable explainations of my concerns, and supplemented necessary ablations. I'm glad to hold my borderline accept for this paper.

---

### Official Review · Reviewer_pCdW · 2025-06-30

**Clarity:** 3
**Significance:** 3
**Originality:** 3
**Rating:** 5
**Confidence:** 4

**Summary:**

This paper aims at solving category-level object pose estimation task, and proposes RFMPose that utilizes Riemannian Flow Matching for pose trajectory alignment. The method is designed upon GenPosee++ framework and proposes a novel matching strategy for precise pose learning. Experiments show the superior performance of this paper.

**Questions:**

please refer to weaknesses.

**Ethical Concerns:**

["NO or VERY MINOR ethics concerns only"]

**Final Justification:**

After rebuttal, I tend to accept this paper.

**Limitations:**

yes

**Paper Formatting Concerns:**

There are no obvious formatting errors in current version.

**Quality:**

3

**Strengths And Weaknesses:**

Strengths:
1. This paper is well organized and written.
2. The novelty is convincing.
3. Experiments are sufficient.

Weaknesses:
1. Basicaly, this is an incremental work that builds RFM into GenPose++ framework and obtain an improvement performance. Although it proposes a new principle and many theorical derivations, personally, I am concerned about its value and novelty. In fact, category-level rigid object pose estimation is a well studied topic, espicially when foundation models, e.g. FoundationPose, are proposed, new types of objects such as garments and transparent objects or new visual tasks such as grasp and manipulation policy are more worthy investigating. Thus, I do not believe that an old visual task with incremental improvement should be accepted. Even though, I give a boardline accept score since this is a technically sound work.
2. Experiments show some object tracking results. But most of baselines are not designed for tracking task, the authors should compare more methods in this table.
3. There are few figures and qualitative results in the main text, making it hard to understand how RFM works and how good results RFMPose obtains.

---

> ### Author Rebuttal · Authors · 2025-07-31
>
> Dear Reviewer:
>
> We sincerely appreciate your acknowledgement of our work, particularly your recognition that our idea is **"well organized and written"** and **"the novelty is convincing"**. We hope to address all your concerns with the following responses:
> _________________
> **Q1. Basicaly, this is an incremental work that builds RFM into GenPose++ framework and obtain an improvement performance. Although it proposes a new principle and many theorical derivations, personally, I am concerned about its value and novelty. In fact, category-level rigid object pose estimation is a well studied topic, espicially when foundation models, e.g. FoundationPose, are proposed, new types of objects such as garments and transparent objects or new visual tasks such as grasp and manipulation policy are more worthy investigating. Thus, I do not believe that an old visual task with incremental improvement should be accepted. Even though, I give a boardline accept score since this is a technically sound work.**
>
> **A1:** We sincerely appreciate your recognition of our work again, specifically your comment that “this is a technically sound work.” We appreciate this opportunity to clarify the value and novelty of our contribution, especially in the context of the important points you have raised.
>
> **First**, we respectfully argue that category-level 6D pose estimation remains a vital research area with significant unsolved challenges for the following two reasons: **1) Limited Generalization in Existing Models**: While category-level rigid object pose estimation has been well-studied,  most of existing methods are validated on benchmarks with a limited number of categories (e.g., 6 classes in NOCS[1] or 10 classes in Housecat6D[2]). These small-scale settings do not fully expose the challenges of high inter-class similarity and large intra-class variation. As demonstrated in our paper, the proposed method obtains superior performance on datasets of great diversity and scale. **2) Impractical Task Setting**: We thank you for mentioning FoundationPose[3], which is indeed a landmark work. However, it is crucial to distinguish its instance-level task from our category-level one. FoundationPose[3] requires a specific 3D CAD model or a set of reference 2D images for each object instance before inference. This prerequisite is often impractical for real-world applications where systems must handle novel, unseen objects within a known category.
>
> **Second**, we completely agree with your insightful comment that "garments and transparent objects or new visual tasks such as grasp and manipulation policy are more worth investigating". In fact, these are research avenues we are actively exploring. Nevertheless, we maintain that 6D object pose estimation remains a fundamental perception module, critical for enabling many of those downstream manipulation tasks, such as [4][5]. Notably, the generative architecture of our approach enables  seamless integration into contemporary diffusion-based Vision-Language-Action (VLA) frameworks, offering superior flexibility for downstream manipulation tasks.
>
> We hope our response clarifies the value of our contribution and its relevance to the broader goals of the computer vision and robotics community. Thank you again for your constructive feedback.
>
> [1] Normalized Object Coordinate Space for Category-Level 6D Object Pose and Size Estimation, CVPR 2019.
>
> [2] HouseCat6D - A Large-Scale Multi-Modal Category Level 6D Object Perception Dataset with Household Objects in Realistic Scenarios, CVPR 2024.
>
> [3] FoundationPose: Unified 6D Pose Estimation and Tracking of Novel Objects, CVPR 2024.
>
> [4] VTAO-BiManip: Masked Visual-Tactile-Action Pre-training with Object Understanding for Bimanual Dexterous Manipulation, IROS 2025.
>
> [5] 6D Object Pose Tracking in Internet Videos for Robotic Manipulation, ICLR 2025.
> _________________
> **Q2. Experiments show some object tracking results. But most of baselines are not designed for tracking task, the authors should compare more methods in this table.**
>
> **A2:** Thank you for your kind suggestions.
>
> **First**, we would like to clarify that our baseline selection was the result of a thorough literature review. To ensure a comprehensive comparison, we referred to the latest survey paper [6] and recent tracking works [8][10] to identify four prominent category-level 6D pose tracking methods published after 2021: BundleTrack[7], CATRE[8], CAPTRA[9], and GenPose++[10].  The only method from this list not included in our table is CAPTRA[9], as its performance is reported to be on par with CATRE[8] (see Table 4 in [8]), which is already one of our key baselines. Therefore, we believe that our selected baselines encompass the most significant and impactful algorithms from top-tier conferences in recent years.
>
> **Second**, we would like to emphasize that our method achieves competitive tracking performance without any dedicated modules or losses designed for tracking, rather than focusing on tracking performance itself. This inherent versatility is a core strength of our approach, showcasing its ability to adapt to downstream tasks like pose tracking with minimal modification. When compared to GenPose++[10], a model that also possesses the aforementioned advantages, our method demonstrates clear superiority in both performance and efficiency.
>
> [6] Deep Learning-Based Object Pose Estimation: A Comprehensive Survey, arXiv:2405.07801
>
> [7] BundleTrack: 6D Pose **Tracking** for Novel Objects without Instance or Category-Level 3D Models, IROS 2021.
>
> [8] CATRE: Iterative Point Clouds Alignment for Category-level Object Pose Refinement, ECCV 2022.
>
> [9] CAPTRA: CAtegory-level Pose **Tracking** for Rigid and Articulated Objects from Point Clouds, ICCV 2021.
>
> [10] Omni6DPose: A Benchmark and Model for Universal 6D Object Pose Estimation and **Tracking**, ECCV 2024.
> _________________
> **Q3. There are few figures and qualitative results in the main text, making it hard to understand how RFM works and how good results RFMPose obtains.**
>
> **A3:** Given the page limits of NeurIPS submission, we have to include visualizations for qualitative analysis and implementation details in the supplementary material. Please refer to the supplementary material for the qualitative visualization results.
>
> As for few figures to illustrate how RFM works, we would like to highlight Figure 2 and Figure 1(b). Specifically, Figure 2 illustrates the overall process of Riemannian Flow Matching for pose estimation, showing the transformation from a random Gaussian pose to the target pose. In contrast, Figure 1(b) details the step-by-step process within the RFM framework, including velocity field matching, likelihood estimation, and supervision. Due to the non-linear nature of SE(3), RFM learns a velocity vector field  in Riemannian space that governs the evolution of poses, endowed with explicit physical interpretability. This enables the enforcement of geodesic constraints on the Riemannian manifold, thereby ensuring geometric consistency throughout the pose generation process. We hope that the explanation above, along with the discussion in Section 3.5, addresses your concern regarding how RFM works.
>
> Thank you for your valuable comment. We will update our schematic diagram and add a paragraph at the beginning of Section 3 in revised manuscript to further clarify the overall pipeline and how our method works.  Additionally, we will revise our manuscript by carefully condensing certain sections to incorporate representative visualizations from the supplementary material into the main manuscript.

---

> > ### Comment · Reviewer_pCdW · 2025-08-05
> > **agree with response and tend to accept**
> >
> > Thanks for your reply. I agree with that rigid pose estimation is valueble and convincing, but I still want to know more details on how to utilize the pose estimation results in manipulation tasks (not simply align the gripper with the 3D bounding box).
> > The authors have to promise that codes will be made open-source (no more than three months after acceptance), and I will suggest to accept.

---

> > > ### Author Response · Authors · 2025-08-05
> > >
> > > Dear Reviewer,
> > >
> > > Thank you for your reply and valuable feedback. We appreciate your recognition of the importance of the 6D rigid pose estimation task and your positive assessment of our work. We also appreciate your inclination to accept our paper.
> > >
> > > To address your concern, we would like to elaborate on the utilization of pose estimation results in manipulation tasks, with two concrete examples from recent literature [4][5]:
> > >
> > > **Using 6D Pose as a Pre-training Objective:** In [4], the authors proposed reconstructing or predicting an object's 6D pose from masked multimodal inputs (e.g., vision, tactile, action, 6D pose) during a pre-training stage. By this pre-training, the model is forced to learn a deep understanding of the object's spatial properties and geometric relationships between hand and objects, which goes beyond simply aligning the gripper with a 3D bounding box.
> > >
> > > **Extracting 6D Pose Trajectories from Videos for Imitation:** In [5], the authors focus on extracting a temporally consistent and smooth 6D pose trajectory of an object being manipulated in a video. This captures the complete dynamic process of the task, rather than just a single static pose, which enables a robot to imitate the fine-grained motions demonstrated by humans in videos.
> > >
> > > We hope this clarifies the advanced applications and broader impacts of 6D pose estimation in manipulation tasks. We commit to making our code publicly available within three months after officially acceptance.
> > >
> > > [4] VTAO-BiManip: Masked Visual-Tactile-Action Pre-training with Object Understanding for Bimanual Dexterous Manipulation, IROS 2025.
> > >
> > > [5] 6D Object Pose Tracking in Internet Videos for Robotic Manipulation, ICLR 2025.

---

### Official Review · Reviewer_gw9Y · 2025-07-02

**Clarity:** 2
**Significance:** 3
**Originality:** 3
**Rating:** 4
**Confidence:** 5

**Summary:**

The paper introduces a new method for category-level object pose estimation using Riemannian Flow Matching. A conditional probability distribution of object pose is learned, which is conditioned on sensory input. Flow matching in SE(3) space is leveraged to learn this distribution. The method also considers symmetry of objects using multiple feasible ground truth poses. Experiments are conducted on the Omni6DPose dataset to verify the effectiveness of the proposed method.

**Questions:**

Please see weaknesses above regarding questions.

**Ethical Concerns:**

["NO or VERY MINOR ethics concerns only"]

**Final Justification:**

The authors claim to improve the paper according to their rebuttal.

**Limitations:**

Yes

**Quality:**

3

**Strengths And Weaknesses:**

Positives:

-	The idea of using Riemannian Flow Matching for category-level object pose estimation is new and novel.

-	The proposed method demonstrates improvements over existing methods on the Omni6DPose dataset.

Negatives:

-	The paper writing needs to be improved by incorporating more details towards how to do object pose estimation using the Riemannian Flow Matching. Most of the description in the paper is about the theory side of the method. Details about the specific problem of object pose estimation are missing. For example, how to process images, how to prepare the dataset for learning, how to define symmetry of objects in the dataset, how to perform inference in details to obtain the final object poses.

-	In another word, an algorithm-level description of how to use the method for object pose estimation is missing in the paper. Most description is about flow matching in the SE(3) space, which can be independent of the pose estimation problem.

-	The paper does not consider shape variation in category-level object pose estimation. Some previous works jointly estimate the object poses and shapes.

-	Why NOCS got all zeros in table 2?

-	No qualitative visualization of the method provided in the paper.

---

> ### Author Rebuttal · Authors · 2025-07-31
>
> Dear Reviewer:
>
> We sincerely appreciate your acknowledgement of our work, particularly your recognition that our idea is **"new and novel"**. We hope to address all your concerns with the following responses:
> _________________
> **Q1. The paper writing needs to be improved by incorporating more details towards how to do object pose estimation using the Riemannian Flow Matching. Most of the description in the paper is about the theory side of the method. Details about the specific problem of object pose estimation are missing. For example, how to process images, how to prepare the dataset for learning, how to define symmetry of objects in the dataset, how to perform inference in details to obtain the final object poses.**
>
> **Q2. In another word, an algorithm-level description of how to use the method for object pose estimation is missing in the paper. Most description is about flow matching in the SE(3) space, which can be independent of the pose estimation problem.**
>
> **A1&A2:** We sincerely appreciate your constructive feedback. Given that the two aforementioned questions are closely related, we therefore respond to them together.
>
>  Due to the strict page limits of the NeurIPS submission, we had to move **Implementation Details** that are not directly germane to the core innovations of our methodology to the supplementary material. We kindly refer you to “Section A. More Implementation Details” in the supplementary material for these details, where we have already elaborated on: 1) How we process images; 2) How we design the network architectures; 3) How we define the symmetry of objects; 4) How we perform training and inference to obtain the final object poses.
>
> Furthermore, we completely agree  the importance of clarifying the overall workflow and algorithm-level descriptions in the main manuscript. To enhance readability, we will revise our manuscript to include:
>
> **1)Overall Pipeline Introduction:** We will add a concise introductory paragraph at the beginning of the “**Section 3 Methodology”** in the revised manuscript that provides an overarching introduction to our entire pose estimation pipeline.
>
> **2) Algorithm-Level Description:** Crucially, to directly address your suggestion, we will incorporate some content from the supplementary material, such as implementation details and visualizations, into the main manuscript, while refining the theoretical introduction in the Preliminaries section.
>
> **3) Enhanced Cross-Referencing:** We will enrich the main manuscript with cross-references to the **Implementation Details** in the supplementary material, ensuring readers can easily access detailed algorithm-level descriptions in the supplementary material.
>
> We believe these revisions will ensure readers can readily grasp the methodological framework, stay focused on the core contributions and novelties, and easily access detailed descriptions when they wish to delve deeper. Thank you again for helping us strengthen our manuscript.
> _________________
> **Q3. The paper does not consider shape variation in category-level object pose estimation. Some previous works jointly estimate the object poses and shapes.**
>
> **A3:** Thank you for this valuable suggestion. Our work focuses on the category-level pose estimation task without joint shape modeling, which aligns with the current mainstream research in this field.   As listed in a recent survey[1], the vast majority of works also concentrate exclusively on the pose estimation task and it is still a distinct challenge.
>
> Meanwhile, we agree that jointly estimating pose and shape is a promising research direction with broad application potential. Given that single-image 3D shape reconstruction has evolved into a well-established field that is now dominated by generative models, this presents an opportune moment to integrate these two domains.
>
> We greatly appreciate your insightful suggestion and concur that joint modeling is a compelling research topic for future exploration, which we will take into serious consideration in our future research.
>
> [1] Deep Learning-Based Object Pose Estimation: A Comprehensive Survey, arXiv:2405.07801.
> _________________
> **Q4. Why NOCS got all zeros in table 2?**
>
> **A4:** Thank you for highlighting this point.  The NOCS methodology operates by first predefining a canonical coordinate space for each object category. It then estimates the pose by predicting a NOCS map from an input image and aligning this map with the corresponding canonical template. While this approach is effective for a small number of categories, such as the six classes in the original NOCS dataset, it faces significant challenges on large-scale and more complex datasets like Omni6DPose. This performance collapse is consistent with the findings in the GenPose++[2], which also reported NOCS's failure on this benchmark but did not provide a detailed analysis of the underlying causes. Here, we would like to elaborate on the two primary reasons for NOCS's failure:
>
> **First**, the scalability of NOCS is limited. When applied to the 140+ classes in our manuscript, the high inter-class visual similarity and large intra-class shape variation become particularly pronounced. Consequently, the NOCS struggles to reliably infer the correct canonical shape from RGB information alone, leading to severe mismatches between the predicted NOCS map and the actual object geometry.
>
> **Second**, the NOCS method is inherently sensitive to visual domain shifts, such as variations in texture and lighting. Our evaluation protocol, where all baseline models are trained on the simulated SOPE dataset and tested on the real-world ROPE dataset, exposes this vulnerability. This significant sim-to-real gap exacerbates the fragility of the NOCS approach, amplifying prediction errors when the model confronts the diverse and challenging conditions of real-world scenarios.
>
> [2] Omni6DPose: A Benchmark and Model for Universal 6D Object Pose Estimation and Tracking, ECCV 2024.
> _________________
> **Q5. No qualitative visualization of the method provided in the paper.**
>
> **A5:** Thanks for your question. Due to the NeurIPS page limit, we have to include the visualizations for qualitative analysis in the supplementary material. We kindly ask you to review  our supplementary material for the qualitative visualization results. Accordingly, we will revise our manuscript by carefully condensing certain sections to incorporate representative visualizations from the supplementary material into the main manuscript.

---

### Official Review · Reviewer_wpVz · 2025-07-03

**Clarity:** 3
**Significance:** 3
**Originality:** 3
**Rating:** 5
**Confidence:** 4

**Summary:**

This paper introduces a framework for category-level object pose estimation by adapting the flow matching paradigm to the non-Euclidean geometry of pose space. The core contribution is the development of a Riemannian Flow Matching process, which correctly operates on the manifold of rotations (SO(3)). To address key challenges within this framework, the authors make two specific technical contributions: (1) They propose a Riemannian Optimal Transport loss designed to effectively learn the multi-modal distributions associated with symmetric objects, a long-standing problem in pose estimation. (2) They incorporate a Likelihood Estimation method to intelligently fuse multiple pose candidates generated by the flow matching process, enhancing the robustness of the final prediction. The effectiveness of this comprehensive framework is validated through experiments on the challenging Omni6DPose dataset, where the proposed method is shown to achieve superior performance for category-level object pose estimation.

**Questions:**

1. In the context of category-level object pose estimation, how to estimate object size under the proposed Riemannian flow matching framework?

2. How about the performance under different levels of occlusions?

3. Under what circumstances that the proposed Riemannian flow matching would exhibit obvious advantages?

**Ethical Concerns:**

["NO or VERY MINOR ethics concerns only"]

**Final Justification:**

The rebuttal has resolved my concerns about the object size calculation, robustness to occlusion, and the advantageous application scenarios for the proposed Riemannian flow matching algorithm. Therefore, I am willing to keep my original score of 'Accept'. Meanwhile, I highly recommend that the authors integrate these clarifications into the final manuscript to improve its completeness and impact.

**Limitations:**

Yes.

**Quality:**

3

**Strengths And Weaknesses:**

Pros:

+ The proposed method is well-motivated, and the proposed Riemannian flow matching framework is well-designed and self-contained.

+ Experiments on the Omni6DPose dataset that covers over 140 categories demonstrate the state-of-the-art performance for category-level object pose estimation.

+ Comparison with conventional Euclidean flow matching demonstrates the effectiveness of  Riemannian flow matching for the task of object pose estimation.

+ The effectiveness of Riemannian Optimal Transfer is also validated via an ablation study.

Cons:

- It is unclear how to compute the object size, given the proposed Riemannian flow matching framework. It is expected to have a description for this part. What does "we used the same scale estimation results as GenPose++" mean? How about the accuracy for the object size estimation?

- Apart from the object symmetry, occlusion is another important concern for an object pose estimation model. I am wondering about the performance of the proposed flow matching method under different levels of occlusions. In addition, I am also wondering about the accuracy of the likelihood estimation method under occlusions.

- A qualitative comparison between the Euclidean flow matching and the proposed Riemannian flow matching would be appreciated. Under what circumstances that the proposed Riemannian flow matching would exhibit obvious advantages?

---

> ### Author Rebuttal · Authors · 2025-07-31
>
> Dear Reviewer:
>
> We sincerely appreciate your acknowledgement of our work, particularly your recognition that the proposed method is **"well-motivated”** **"well-designed"** and **"self-contained"**. We hope address all your concerns with following responses:
>
> _________________
> **Q1. It is unclear how to compute the object size, given the proposed Riemannian flow matching framework. It is expected to have a description for this part. What does "we used the same scale estimation results as GenPose++" mean? How about the accuracy for the object size estimation?**
>
> **A1:** Thanks for bringing it up. We would like to clarify that the object size estimation in our method builds on the implementation in GenPose++, which employs an extra Scale Network to predict the object's scale using concatenated features from the RGB image and point cloud. This Scale Network features a very simple architecture with three linear layers and, notably, is trained independently of the main pose estimation networks. To ensure a fair comparison, we employ the same standalone Scale Network and weights as GenPose++ for scale estimation, thereby isolating our model's performance from potential variations in scale prediction.
>
> To evaluate the accuracy of object size estimation, we performed an evaluation with the object's center point and rotation fixed. This evaluation yielded a mean 3D Intersection over Union (3D IoU) of **67.61%** for 3D bounding boxes.
>
> _________________
> **Q2. Apart from the object symmetry, occlusion is another important concern for an object pose estimation model. I am wondering about the performance of the proposed flow matching method under different levels of occlusions. In addition, I am also wondering about the accuracy of the likelihood estimation method under occlusions.**
>
> **A2:**  Thank you for your insightful feedback.  The Omni6DPose[1] dataset does not include occlusions in training and testing. Our work and previous works like GenPose++ and GenPose do not consider the occlusion issue. As during training, the networks do not see objects under occlusions, these works may not work robustly under occlusions.
>
> Despite that,  we agree that studying the performance under occlusion is important, and we designed an experiment to evaluate robustness. We introduced random square occluders covering 10% of the object’s RGB image area and correspondingly removed the depth information from the occluded regions in the point cloud.
>
> Under this 10% random occlusion setting, our model achieves a performance of **40.9 (-7.1)** on $IoU_{25}$ and **26.0 (-5.6)** on $10^{∘}5cm$. For comparison, GenPose++[1] achieves **36.8 (-7.1)** on $IoU_{25}$ and **23.8 (-4.7)** on $10^{∘}5cm$. These results indicate that occlusion exerts a notable performance degradation on the 6D pose estimation performance for both our model and GenPose++. We primarily attribute this to the complete absence of occluded instances in the training set.
>
> To further assess the impact of occlusion on our likelihood estimation, we performed an ablation study. Disabling the likelihood module resulted in a performance of 39.8 on $\mathbf{IoU}_{25}$, 8.1 on $5^{∘}2cm$, and 12.2 on $5^{∘}5cm$. In comparison, with likelihood estimation enabled, the performance improves to 40.9, 9.3, and 12.8, yielding performance gains of **+1.1**, **+1.2**, and **+0.6** on the respective metrics. These findings demonstrate that the proposed likelihood estimation remains effective even in the presence of occlusion.
>
> Thank you for your constructive suggestion, and we will incorporate these experimental results into our revised supplementary material.
>
> [1]  Omni6DPose: A Benchmark and Model for Universal 6D Object Pose Estimation and Tracking, ECCV 2024.
>
> _________________
> **Q3. A qualitative comparison between the Euclidean flow matching and the proposed Riemannian flow matching would be appreciated. Under what circumstances that the proposed Riemannian flow matching would exhibit obvious advantages?**
>
> **A3:** Thank you for your valuable feedback. Due to the NeurIPS 2025 rebuttal policy, which prohibits the inclusion of images or external links, we regret that we are unable to provide additional visualization analysis for the ablation studies on Riemannian Flow Matching. However, we can still address the question of when our Riemannian Flow Matching exhibits clear advantages. We conducted a statistical analysis of the performance changes across 145 object categories and found that this method enhances performance in the vast majority of cases, yielding improvements in 82% of categories. These statistical results suggest that our Riemannian Flow Matching is broadly effective in learning object pose distributions, particularly in asymmetric classes (yielding improvements in 90% of categories).

---

> > ### Comment · Reviewer_wpVz · 2025-08-05
> > **Official Comment from Reviewer wpVz**
> >
> > Thanks for the rebuttal, which has resolved my concerns about the object size calculation, robustness to occlusion, and the advantageous application scenarios for the proposed Riemannian flow matching algorithm.

---

### Note · Authors · 2025-08-11

Dear Reviewers and Area Chair,

We would like to express our sincere gratitude to all reviewers for your valuable suggestions and insights.  We appreciate all of you for your comments highlighting the strengths, which we summarize as follows:

**1. Clear motivation** (Reviewers wpVz, gw9Y and iY4w)

**2. Novelty** (Reviewers gw9Y, pCdW, and iY4w)

**3. Rigorous derivations and self-contained presentation** (Reviewers wpVz and iY4w)

**4. Well-organized and well-written** (Reviewer pCdW)

We are glad to hear that our rebuttal successfully addressed the concerns of Reviewer wpVz and Reviewer iY4w. Nonetheless, a major concern most reviewers share is the lack of qualitative comparisons for visualization results in the main manuscript and a more detailed algorithm-level description or schematic diagram. We will address this concern and polish our paper in the revised version. Specifically, we will make the following improvements:
1. We will incorporate representative visualizations from the supplementary material directly into the main body of the paper to better showcase our results;
2. We will revise the schematic diagram and add a paragraph at the beginning of Section 3 to further clarify the overall pipeline and how our method works.

Finally, we would like to express our great appreciation and excitement that all the reviewers recognize our contributions and express a positive assessment of our work. We want to emphasize that our work contributes to the community not only in computer vision but also in robotics. We believe that it is worth publishing to simulate further applications in the future.

---

### Decision · Program_Chairs · 2025-09-17

**Decision:**

Accept (poster)

**Comment:**

The authors propose a Riemannian Flow Matching procedure for pose estimation. It includes a novel optimal transport loss and a method to fuse multiple pose candidates. The reviewers welcomed the well-motivated and novel approach and acknowledged that the approach was well evaluated in experiments and ablation studies.

Before the discussion phase, there were some reservations regarding clarity and the reviewers were asking for qualitative results to validate the approach. Clarity concerns could be resolved while qualitative results could not be shown in the discussion phase. All in all, the reviewers remained with their positive stance on the paper. I follow and recommend to accept this work.